# Pilot Study on Neonatal Screening for Methylmalonic Acidemia Caused by Defects in the Adenosylcobalamin Synthesis Pathway and Homocystinuria Caused by Defects in Homocysteine Remethylation

**DOI:** 10.3390/ijns7030039

**Published:** 2021-07-07

**Authors:** Reiko Kagawa, Go Tajima, Takako Maeda, Fumiaki Sakura, Akari Nakamura-Utsunomiya, Keiichi Hara, Yutaka Nishimura, Miori Yuasa, Yosuke Shigematsu, Hiromi Tanaka, Saki Fujihara, Chiyoko Yoshii, Satoshi Okada

**Affiliations:** 1Department of Pediatrics, Hiroshima University Graduate School of Biomedical and Health Sciences, Minami-ku, Hiroshima 734-8551, Japan; rekagawa@hiroshima-u.ac.jp (R.K.); d185866@hiroshima-u.ac.jp (F.S.); sokada@hiroshima-u.ac.jp (S.O.); 2Division of Neonatal Screening, Research Institute, National Center for Child Health and Development, Setagaya-ku, Tokyo 157-8535, Japan; maeda-t@ncchd.go.jp; 3Department of Pediatrics, Hiroshima Prefectural Hospital, Minami-ku, Hiroshima 734-8530, Japan; a-utsunomiya@hph.pref.hiroshima.jp; 4Department of Pediatrics, National Hospital Organization Kure Medical Center and Chugoku Cancete Center, Kure 737-0023, Japan; hara.keiichi.dv@mail.hosp.go.jp; 5Department of General Perinatology, Hiroshima City Hiroshima Citizens Hospital, Naka-Ku, Hiroshima 730-8518, Japan; warabikinako@gmail.com; 6Department of Pediatrics, Faculty of Medical Sciences, University of Fukui, Eiheiji-cho, Fukui 910-1193, Japan; miori@u-fukui.ac.jp (M.Y.); yosuke@u-fukui.ac.jp (Y.S.); 7Hiroshima City Medical Association Clinical Laboratory, Naka-ku, Hiroshima 730-8611, Japan; saiboushin@labo.city.hiroshima.med.or.jp (H.T.); sententaisha@labo.city.hiroshima.med.or.jp (S.F.); yoshii@labo.city.hiroshima.med.or.jp (C.Y.)

**Keywords:** neonatal screening, homocystinuria, methylmalonic acidemia, disorders of cobalamin metabolism, hypomethioninemia

## Abstract

Neonatal screening (NS) for methylmalonic acidemia uses propionylcarnitine (C3) as a primary index, which is insufficiently sensitive at detecting methylmalonic acidemia caused by defects in the adenosylcobalamin synthesis pathway. Moreover, homocystinuria from cystathionine β-synthase deficiency is screened by detecting hypermethioninemia, but methionine levels decrease in homocystinuria caused by defects in homocysteine remethylation. To establish NS detection of methylmalonic acidemia and homocystinuria of these subtypes, we evaluated the utility of indices (1) C3 ≥ 3.6 μmol/L and C3/acetylcarnitine (C2) ≥ 0.23, (2) C3/methionine ≥ 0.25, and (3) methionine < 10 μmol/L, by retrospectively applying them to NS data of 59,207 newborns. We found positive results in 116 subjects for index (1), 37 for (2), and 15 for (3). Second-tier tests revealed that for index 1, methylmalonate (MMA) was elevated in two cases, and MMA and total homocysteine (tHcy) were elevated in two cases; for index 2 that MMA was elevated in one case; and for index 3 that tHcy was elevated in one case. Though data were anonymized, two cases identified by index 1 had been diagnosed with maternal vitamin B_12_ deficiency during NS. Methylene tetrahydrofolate reductase deficiency was confirmed for the case identified by index 3, which was examined because an elder sibling was affected by the same disease. Based on these data, a prospective NS study is underway.

## 1. Introduction

Current neonatal screening (NS) in Japan identifies methylmalonic acidemia and propionic acidemia by elevated levels of propionylcarnitine (C3), and homocystinuria caused from cystathionine β-synthase (CBS) deficiency by elevated levels of methionine (Met). However, C3 is not always sufficiently sensitive to detect methylmalonic acidemia caused by defects in the adenosylcobalamin synthesis pathway, as we show below in a case of cobalamin D disease (cblD) variant 2 missed in NS. Moreover, Met levels actually decrease in homocystinuria resulting from defects in homocysteine remethylation. The prognosis of these diseases can be greatly improved by starting specific medication in the early neonatal period [1,2,3,4,5,6,7], as observed through the management of two siblings affected by methylenetetrahydrofolate reductase (MTHFR) deficiency, which is described below.

## 2. Materials and Methods

### 2.1. Preliminary Retrospective Study

In Japan, dried blood spot (DBS) testing for NS generally takes place on the fourth or fifth day after birth. In the Hiroshima area, there are approximately 20,000 births each year, and all NS samples are analyzed in the Hiroshima City Medical Association Clinical Laboratory. To improve the sensitivity of current NS for methylmalonic acidemia caused by defects in the adenosylcobalamin synthesis pathway and to establish NS for homocystinuria caused by defects in homocysteine remethylation, we planned a preliminary retrospective study to apply the following indices to NS data from April 2015 to September 2017: (1) C3 and C3/acetylcarnitine (C2) (current NS indices for methylmalonic acidemia and propionic acidemia), (2) C3/Met, and (3) Met (the lower cutoff).

### 2.2. Prospective Pilot Study

After evaluating positive rates for each index, we enrolled newborns from 10 major hospitals in the Hiroshima area into a pilot study on prospective NS. Parents provided their written informed consent for participation. Samples were anonymized by the removal of personal information. Samples that met one or more of the three indices were transported to the National Center for Child Health and Development for the second-tier measurement of methylmalonate (MMA) and total homocysteine (tHcy). Patients with elevated MMA and/or tHcy were further examined in the Department of Pediatrics, Hiroshima University Hospital.

### 2.3. Biochemical Analysis

Analysis of amino acids and acylcarnitines in the NS DBS was performed using the flow injection method with an LCMS-8030 tandem mass spectrometer (Shimadzu, Kyoto, Japan). The second-tier measurement of MMA and tHcy in DBS was performed using liquid chromatography–mass spectrometry with an LCMS-8040 tandem mass spectrometer (Shimadzu). Cutoff values for MMA and tHcy were 1 μmol/L and 5 μmol/L, respectively.

### 2.4. Statistical Analysis

Statistical analyses of NS test results were performed using a Tandem Internal Quality Control System (System Kay, Kyoto, Japan) and Histogram Creation Sheet (Technical Subcommittee, Japanese Society for Neonatal Screening, Tokyo, Japan).

## 3. Case Report

### 3.1. Case 1

A baby boy born as the first child of healthy nonconsanguineous parents at 37 weeks’ gestation weighed 2864 g at birth. His NS DBS showed that C3 level was elevated to 4.79 nmol/mL (cutoff, 3.6 nmol/mL), but C3/C2 was considered normal at 0.231 (cutoff, 0.25). He showed normal growth and psychomotor development. At 1 year of age, he had norovirus gastroenteritis, presenting with vomiting, groaning, and impaired consciousness, and was taken to an emergency hospital. Blood tests revealed marked acidosis, and plasma ammonia was elevated to 251 μg/dL (normal range, 30–80 μg/dL). Further diagnostic analysis revealed plasma MMA levels of 132.7 nmol/mL (normal range, 0.23–0.45 nmol/mL), suggestive of MMA. Lymphocyte methylmalonyl-CoA mutase activity was normal in the presence of adenosylcobalamin (56.2 pmol succinyl-CoA/min/10^6^ cells, control 61.6 ± 22.2). His serum tHcy and vitamin B_12_ levels were normal. Based on these results, he was diagnosed with suspected vitamin B_12_-responsive methylmalonic acidemia. A vitamin B_12_ challenge test was performed by daily infusion of 1 mg cyanocobalamin for 5 days. Post-challenge, his MMA levels decreased. Genetic analysis revealed compound heterozygous variants in *MMADHC*; c.18T > A (p.C6X) and c.702insT. Based on a diagnosis of CblD (variant 2), cobalamin and carnitine therapy was started. This case was reported previously [8].

### 3.2. Case 2

A baby girl born as the first child of healthy nonconsanguineous parents at 40 weeks’ gestation weighed 2810 g at birth, and NS results were normal. However, her sucking was weak and her weight gain was poor. From 2.5 months of age, she presented with hypertonia and the setting-sun eye phenomenon. Although ultrasonography of her brain at 13 days old showed no abnormal findings (Figure 1), head magnetic resonance imaging (MRI) at 2.5 months revealed marked ventricular enlargement, suggesting hydrocephalus or brain atrophy (Figure 2). She underwent ventricular drainage, but respiratory failure became evident at 4 months of age when there was no improvement in head MRI findings. Further diagnostic analysis revealed plasma tHcy levels of 170 μmol/L (normal range, 3.7–13.5 μmol/L) and urinary Hcy levels of 510 μmol/mg·cre (reference value, “undetectable”), suggestive of homocystinuria. As plasma Met level was as low as 3.4 μmol/L (normal range, 18.9–40.5 μmol/L), defects in homocysteine remethylation were indicated. A Met decrease in the NS DBS was also ascertained retrospectively (6.6 μmol/L).

The administration of betaine monohydrate (100 mg/kg/day) was started at 4 months of age, and her respiratory status and vitality improved rapidly. Sanger sequencing of the methylenetetrahydrofolate reductase gene (*MTHFR*) detected a homozygous variant, c.466_467GC > TT, and both parents were found to be heterozygous carriers of this variant. Based on the diagnosis of homocystinuria type III caused by MTHFR deficiency, betaine therapy was continued at the dosage of 300 mg/kg/day, which raised plasma Met levels to 14–40 μmol/L, and reduced plasma tHcy concentrations to 50–110 μmol/L. Head MRI at the age of 12 months revealed the almost complete resolution of ventricular enlargement and atrophic changes (Figure 3). However, severe psychomotor retardation became evident, with a development quotient of 36 at the age of 1 year and 4 months. Epileptic seizures also appeared at the age of 3 years, so the administration of sodium valproate was added. This case was reported previously [9].

### 3.3. Case 3

A baby boy born as the second child of the same parents as Case 2 at the gestational age of 38 weeks and 5 days, with a birth weight of 2936 g, had a normal perinatal course. Due to the medical history of his sister (Case 2), blood samples were collected within 24 h of birth. Concentrations of Met in his DBS and plasma were 8.7 μmol/L and 5.4 μmol/L, respectively. Plasma tHcy levels were elevated to 97.4 μmol/L, which was associated with increased urinary Hcy levels (3437.1 μmol/mg·cre). These data suggested MTHFR deficiency, so the administration of betaine at 300 mg/day (approximately 100 mg/kg/day) was started at the age of 7 days. Thereafter, plasma Met and tHcy were controlled within the range of 12–15 μmol/L and 80–120 μmol/L, respectively. He has maintained normal growth and psychomotor development up to the age of 12 months, and no abnormalities were found on head MRI. This case was reported previously [9].

## 4. Results

### 4.1. Preliminary Retrospective Study

Prior to the preliminary retrospective study, we evaluated statistical data for C3, C3/C2, Met, and C3/Met in the DBS of 23,467 newborns in the Hiroshima area from April 2016 to March 2017 (Table 1 and Table 2). The C3 cutoff has remained at 3.6 μmol/L since the start of tandem mass spectrometry (MS/MS)-based NS in 2013. This value corresponds to the 98.1st percentile of the enrolled data, resulting in a positive rate of 1.82%. The C3/C2 cutoff needs adjusting every few years, so was set at 0.22. The combination of cutoffs for C3 and C3/C2 yielded a positive rate of 0.09%. The 99.9th percentile and 99.5th percentile of C3/Met were 0.25 and 0.20, respectively. Setting the C3/Met cutoff at 0.25 (the 99.9th percentile) gave a positive rate of 0.13%, which was appropriate for the first screening test. However, a lower Met cutoff was required to detect MTHFR deficiency. Cutoffs of 9.0 μmol/L and 10.0 μmol/L achieved positive rates of 0.05% and 0.12%, respectively. Based on these data, we established the following cutoffs for the preliminary study: (1) C3 ≥ 3.6 μmol/L and C3/C2 ≥ 0.22, (2) C3/Met ≥ 0.25, and (3) Met < 10.0 μmol/L.

For the preliminary study, NS data of 59,207 newborns were evaluated, and a total of 116, 37, and 15 newborns were selected for second-tier tests using indices 1–3, respectively. For index 1, we observed a MMA increase in two cases, and increased MMA and tHcy in two cases. For index 2, we observed a MMA increase in one case. For index 3, we observed a tHcy increase in one case (Table 3). Though further examination was not included in this study, three out of the four cases assessed using index 1 were shown to be positive for screening with C3 and C3/C2 indicators in the current NS, and maternal vitamin B_12_ deficiency was confirmed in two of them. One case with increased MMA had no apparent cause. The case with increased tHcy measured using index 3 was Case 3 described above.

### 4.2. Prospective Pilot Study

Between April 2019 and December 2020, 6080 of 40,595 newborns in the Hiroshima area were enrolled in the pilot study. The C3/C2 cutoff is reviewed every few years, and was set at 0.23 from April 2019 (data not shown). Therefore, we set the C3/C2 cutoff to 0.23 in the prospective pilot study. The number of cases shown to be positive was two using index 1 alone, one using both indices 1 and 2, seven using index 2 alone, eight using indices 2 and 3, and 54 using index 3 alone (Table 4). Only one subject out of a total of 72 with a positive finding had increased MMA levels in the second-tier tests, but no increase in serum MMA or plasma tHcy was observed on detailed examination (data not shown). Additionally, no obvious pathological variants were detected in genes associated with cobalamin metabolism (*MMAA*, *MMAB*, *MMACHC*, *MMADHC*), *MUT*, *PCCA*, or *PCCB* (data not shown).

To investigate the low Met levels of low birth weight infants, each index was examined in the NS carried out from April to October 2019 for infants with a birth weight of ≤2000 g who were part of the pilot study. No association with low birth weight infants was found for index 1, but the frequency of low birth weight infants increased for indices 2 and 3 (Table 5).

## 5. Discussion

Fifteen years after the pilot study in 1997, MS/MS-based NS was adopted as an official Japanese public health care service in 2013 [11]. Its target diseases include methylmalonic acidemia and homocystinuria caused by CBS deficiency. In clinical practice, however, we encountered two symptomatic infants with biochemical profiles identical to those of methylmalonic acidemia who were diagnosed with cblD variant 2 (Case 1) and maternal vitamin B_12_ deficiency (data not shown), respectively, and their NS results were within normal range. Retrospective evaluation of NS data from the cblD patient revealed mild elevation of C3 with a C3/C2 value slightly below the cutoff. The first NS test for both C3 and C3/C2 had been positive in the patient with maternal vitamin B_12_ deficiency, but their second NS test was normal. Additionally, several previous studies reported that methylmalonic acidemia caused by defects in the adenosylcobalamin synthesis pathway tend to show only a slight increase in C3, if any, in neonatal DBS [1,2,11,12,13]. As the symptoms of some of these patients can easily be prevented by the specific administration of vitamin B_12_, more sensitive NS tests are required to enable medication to be administered before the clinical onset of disease [1,2,3,11,12,13,14].

In the present study, we document our experience of two siblings with MTHFR deficiency who followed contrasting clinical courses. The differences in their prognoses appear to be dependent upon the timing when betaine therapy was started. Betaine (*N*,*N*,*N*-trimethylglycine) is the substrate for betaine-homocysteine methyltransferase (BHMT) and thus serves as a methyl donor instead of methylcobalamin. Though the physiological function of BHMT cannot compensate for methionine synthase which requires methylcobalamin, the pharmacological dosage of betaine is effective in reducing tHcy and increasing Met levels in the blood. Met is converted into S-adenosylmethionine (SAM), which is an important methyl donor for various methylation reactions. Therefore, maintaining normal levels of plasma Met is essential in preventing SAM deficiency, which causes severe damage to the central nervous system, especially during infancy and childhood. As it has been shown that the early introduction of betaine therapy can suppress the symptoms of homocystinuria caused by remethylation defects [1,2,3,4,5,6,7], a highly preventive effect of NS is expected. In our preliminary retrospective study, the Met cutoffs of 9.0 μmol/L and 10.0 μmol/L achieved positive rates of 0.05% and 0.12%, respectively, and none of the newborns had Met levels below 8.0 μmol/L. Referring to the cases of MTHFR deficiency that we experienced (Case 2) and previously reported cases, we set the Met cutoff at 10 μmol/L to perform more sensitive NS.

Several studies have been conducted on primary indices and second-tier tests to determine if they have sufficient sensitivity and specificity for screening for cobalamin metabolic disorders [2,10,12,13,15,16,17]. MMA and tHcy in DBS are recommended as promising metabolites for the second-tier measurement [1,2], but current NS practice for these diseases varies between countries [2]. In Japan, our pilot study is the first known trial of prospective screening. In the prospective pilot study, using indices C3, C3/C2, C3/Met, and Met with a lower cutoff increased the number of newborn babies targeted for the second-tier test to 1.18% (72 out of 6080 newborns). By combining the measurement of MMA and tHcy as a second-tier test, only one newborn was found to have an elevated MMA level. Measuring MMA and tHcy as a second-tier is apparently useful in reducing false positives.

Our prospective pilot study raises the question of why the number of newborns with C3/Met levels above the cutoff and below the Met level were higher in the Hiroshima area during the study period. Taking into consideration the fact that newborns enrolled in this study were limited to those born in 10 major hospitals, of which many have a neonatal intensive care unit, and that the ratio of newborns enrolled in this study was as low as 15%, we speculate that the study group has a higher frequency of low birth weight infants than the surrounding area. Low birth weight and preterm infants were previously reported to have low levels of Met concentration in their blood [2], and our NS data suggest similar tendencies (Table 5).

## 6. Conclusions

Although no affected patients has been detected in our prospective pilot study so far, the use of indices C3, C3/C2, C3/Met, and Met with a lower cutoff in combination with second-tier measurements of MMA and tHcy seem to be promising in the establishment of NS for methylmalonic acidemia caused by defects in the adenosylcobalamin synthesis pathway and homocystinuria caused by defects in homocysteine remethylation. We should discuss in this study whether newborns with the real target disease can be detected, and if there are any undetected cases. The progress of future research will be clarified.

## Figures and Tables

**Figure 1 IJNS-07-00039-f001:**
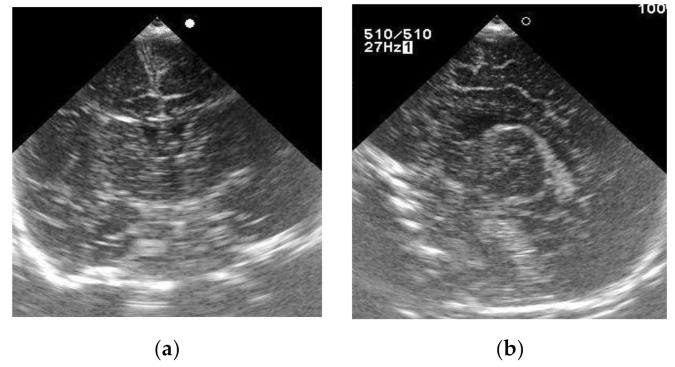
Brain ultrasonography of Case 2 at 13 days of age. (**a**) Coronal plane; (**b**) sagittal plane.

**Figure 2 IJNS-07-00039-f002:**
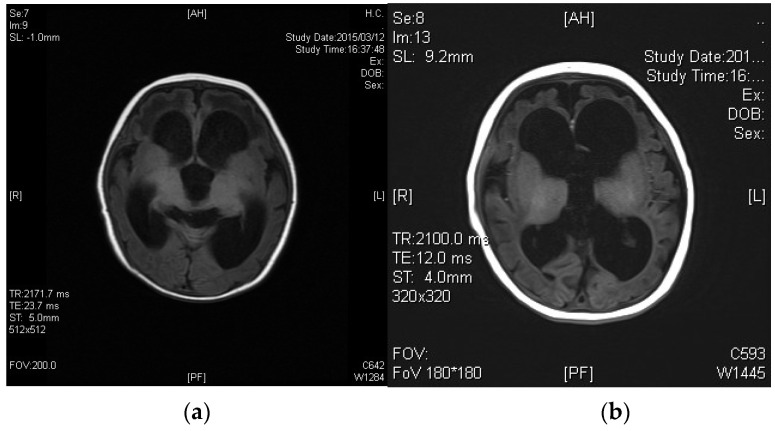
Brain MRI of Case 2. (**a**) At 2.5 months of age (T1); (**b**) 4 months of age (T1) showing no effect on the ventricular size after ventricular drainage.

**Figure 3 IJNS-07-00039-f003:**
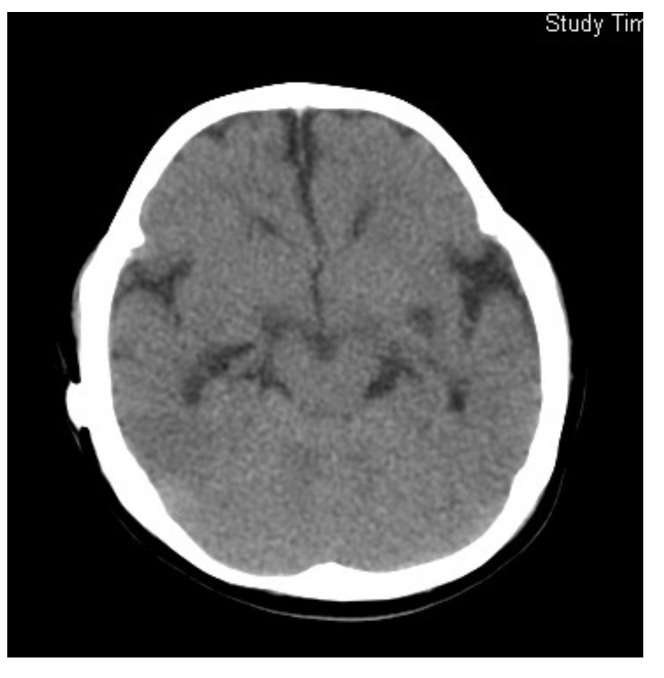
Brain CT of Case 2 at 12 months of age, 7 months after starting betaine treatment.

**Table 1 IJNS-07-00039-t001:** Index distributions of dried blood spots from newborns in the Hiroshima area from April 2016 to March 2017 (*n* = 23,467).

	Mean	99th Centile	99.5th Centile	99.9th Centile
C3 (μmol/L)	1.83	3.96	4.36	5.8
C3/C2	0.09	0.18	0.19	0.24
C3/Met	0.08	0.18	0.20	0.25
Met (μmol/L)	22.10	34.71	36.60	41.75

**Table 2 IJNS-07-00039-t002:** Newborns meeting different cutoff levels in the Hiroshima area from April 2016 to March 2017 (*n* = 23,437).

Index	Cutoff Level					
C3 (μmol/L)	3.5		3.6		3.7	
	n	%	n	%	n	%
	513	2.19	423	1.82	350	1.49
C3/C2^1^	0.22		0.23		0.24	
	n	%	n	%	n	%
	30	0.12	23	0.09	15	0.06
C3/C2 and C3 ≥ 3.6 μmol/L ^1^	0.22		0.23		0.24	
	n	%	n	%	n	%
	22	0.09	16	0.08	13	0.05
C3/Met ^2^	0.24		0.25		0.26	
	n	%	n	%	n	%
	12	0.13	10	0.13	8	0.10
Met (μmol/L)	9		10		11	
	n	%	n	%	n	%
	11	0.05	28	0.12	56	0.24

^1^ Data of 23,390 newborns; ^2^ data of 7714 newborns.

**Table 3 IJNS-07-00039-t003:** Retrospective screening for disorders of cobalamin metabolism in 59,207 newborns in the Hiroshima area from April 2015 to September 2017.

Index	First Test (*n*)	Second-Tier Test (n)			
		Elevated MMA	Elevated MMA and tHcy	Elevated tHcy	Total
(1) C3 ≥ 3.6 μmol/L and C3/C2 ≥ 0.22	116	2	2	0	4
(2) C3/Met > 0.25 and Met < 10 μmol/L	37	1	0	0	1
(3) Met < 10 μmol/L	15	0	0	1	1
Total	168 (0.35%)	3	2	1	6 (3.67%)

Reference range: methylmalonate (MMA) < 1 μmol/L; total homocysteine (tHcy) < 5 μmol/L.

**Table 4 IJNS-07-00039-t004:** Prospective pilot screening in the Hiroshima area from April 2019 to March 2021.

	Newborns Enrolled in This Study (*n* = 6080)	All Newborns in the Area (*n* = 40,595)
Index	First Test, *n* (%)	
(1) C3 ≥ 3.6 μmol/L and C3/C2 ≥ 0.23	3 (0.05)	21 * ^1^ (0.05)
(2) C3/Met > 0.25	15 * ^2^ (0.24)	54 * ^3^ (0.13)
(3) Met < 10 μmol/L	54 (0.89)	271 (0.05)
Total	72 (1.18)	346 (0.85)
	Second test, *n*	
Elevated MMA	1	ND
Elevated tHcy	0	ND

Reference range: methylmalonate (MMA) < 1 μmol/L; total homocysteine (tHcy) < 5 μmol/L. * ^1^ 4 using indices (1) and (2); * ^2^ 8 using indices (2) and (3); * ^3^ 10 using indices (2) and (3).

**Table 5 IJNS-07-00039-t005:** Correlation between low birth weight and hypomethioninemia.

	Birth Weight ≥ 2000 g, *n* (%)	Birth Weight < 2000 g, *n* (%)	*p*-Value
*n* (%)	12,191 (98.32)	209 (1.68)	
(1) C3 ≥ 3.6 μmol/L and C3/C2 ≥ 0.23	5 * ^1^ (0.04)	(0)	–
(2) C3/Met > 0.25	3 (0.02)	3 (1.43)	<0.001
(3) Met < 10 μmol/L	74 (0.61)	26 (12.44)	<0.001

* ^1^ 1 using indices (1) and (2).

## Data Availability

The data that support the findings of this study are available from the corresponding author, Go Tajima.

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
