# Peer review of "Pilot Study on Neonatal Screening for Methylmalonic Acidemia Caused by Defects in the Adenosylcobalamin Synthesis Pathway and Homocystinuria Caused by Defects in Homocysteine Remethylation"

_2409-515X, 2021, doi:10.3390/ijns7030039_

Round 1
Reviewer 1 Report
First of all, the title is misleading because the authors report and discuss not only cobalamin metabolism defects but also MTHFR.
The paper was not written well and it is difficult to follow the author’s discussion points and NBS methods.
The authors wrote that “ however, C3 is not always sufficiently sensitive to detect vitamin B12-responsive methylmalonic acidemia caused by defects in adenosylcobalamin synthesis” Do they have any published paper specific to adenosylcobalamin def. such as CblA or B missed by NBS supporting this sentence. References should be added. It is also not clear why this is important since the authors’ aim to detect low methionine level and related disorders.
The authors report that one case with Cbl D disease had normal NBS. What type of Cbl D disease- D1? or D2? ? How is it related to this study ( hypomethioninemia ?). Adjusting their C3 cut off could be the solution not to miss these cases. It is known that mild forms of Cbl C can be missed since they may not have elevated C3 levels at birth( JIMD Rep. 2015; 23: 71–75).
The authors mix up pathways and their points . I agree that there is need to add total Hcy as a second-tier test or as primary test not to miss patients with classical HCU ( many nbs programs have keep methionine cut off too high to decrease false positives and many cases do not have elevated methionine at 4 days of life). There is also need to detect remthylation defects such as MTHFR , Cbl G or E , and low methionine levels detected on NBS could help to diagnoses these disorders.
The authors also do not discuss false positive rates that would be resulted in by lowering methionine cut off. They should discuss what they suggest to decrease false positives more in detail such as using total Hcy , and report cases out if they have both low methionine and high t Hcy. How about Met/Tyr or Met /Phe ratios. Would they help?
I urge the authors to focus on their aim and discuss their results and outcomes based on the aim of this paper clearly.
Author Response
First of all, the title is misleading because the authors report and discuss not only cobalamin metabolism defects but also MTHFR.
The purpose of our study is to improve the sensitivity of current official screening for methylmalonic acidemia caused by defects in the adenosylcobalamin synthesis pathway, and to detect patients with homocystinuria caused by defects in homocysteine remethylation including MTHFR deficiency. Based on your suggestion, we have changed the title as follows:
Pilot Study on Neonatal Screening for Methylmalonic Acidemia Caused by Defects in the Adenosylcobalamin Synthesis Pathway and Homocystinuria Caused by Defects in Homocysteine Remethylation
The authors wrote that “however, C3 is not always sufficiently sensitive to detect vitamin B12-responsive methylmalonic acidemia caused by defects in adenosylcobalamin synthesis” Do they have any published paper specific to adenosylcobalamin def. such as CblA or B missed by NBS supporting this sentence. References should be added. It is also not clear why this is important since the authors’ aim to detect low methionine level and related disorders.
The authors report that one case with Cbl D disease had normal NBS. What type of Cbl D disease- D1? or D2? ? How is it related to this study ( hypomethioninemia ?). Adjusting their C3 cut off could be the solution not to miss these cases. It is known that mild forms of Cbl C can be missed since they may not have elevated C3 levels at birth( JIMD Rep. 2015; 23: 71–75).
We have added one case of CblD (variant 2) disease to the revised manuscript. This case was previously reported by Ono, H. et al. (J Jap Soc Mass-screening 2014; 24: 49-56 [in Japanese]). This study also reported that cobalamin metabolic disorders caused by decreased MeCbl show a small increase in C3, but a larger C3/Met increase.
The authors mix up pathways and their points. I agree that there is need to add total Hcy as a second-tier test or as primary test not to miss patients with classical HCU (many nbs programs have keep methionine cut off too high to decrease false positives and many cases do not have elevated methionine at 4 days of life). There is also need to detect remthylation defects such as MTHFR , Cbl G or E , and low methionine levels detected on NBS could help to diagnoses these disorders.
Our study aimed to detect methylmalonic acidemia caused by defects in the adenosylcobalamin synthesis pathway, and to detect patients with homocystinuria caused by defects in homocysteine remethylation including MTHFR deficiency. We speculate that tHcy measurement for the second-tier test may be useful in reducing the number of false positives.
The authors also do not discuss false positive rates that would be resulted in by lowering methionine cut off. They should discuss what they suggest to decrease false positives more in detail such as using total Hcy, and report cases out if they have both low methionine and high t Hcy. How about Met/Tyr or Met /Phe ratios. Would they help?
In our preliminary retrospective study, the methionine cutoffs of 9.0 μmol/L and 10.0 μmol/L achieved positive rates of 0.05% and 0.12%, respectively, and none of the newborns had methionine levels below 8.0 μmol/L. Based on our observations of methionine levels in MTHFR dediciency cases of 6.6 μmol/L and 8.7 μmol/L, we set the methionine cutoff value to 10 μmol/L. We did not evaluate Met/Tyr or Met/Phe in the present study.

Reviewer 2 Report
nice work of carefully collected data over many years for validation of new interpretation tools for the remethylation disorders.
Author Response
Thank you for your review comments. I will add more one case report and submit
manuscript again.
